# Is Natural Necessary? Human Voice versus Synthetic Voice for Intelligent Virtual Agents

Amal Abdulrahman [ID] and Deborah Richards *[ID]

School of Computing, Macquarie University, 4 Research Park Dr, Macquarie Park, NSW 2113, Australia; amal.abdulrahman@students.mq.edu.au
* Correspondence: deborah.richards@mq.edu.au

**Abstract:** The use of intelligent virtual agents (IVA) to support humans in social contexts will depend on their social acceptability. Acceptance will be related to the human's perception of the IVAs as well as the IVAs' ability to respond and adapt their conversation appropriately to the human. Adaptation implies computer-generated speech (synthetic speech), such as text-to-speech (TTS). In this paper, we present the results of a study to investigate the effect of voice type (human voice vs. synthetic voice) on two aspects: (1) the IVA's likeability and voice impression in the light of co-presence, and (2) the interaction outcome, including human–agent trust and behavior change intention. The experiment included 118 participants who interacted with either the virtual advisor with TTS or the virtual advisor with human voice to gain tips for reducing their study stress. Participants in this study found the voice of the virtual advisor with TTS to be more eerie, but they rated both agents, with recorded voice and with TTS, similarly in terms of likeability. They further showed a similar attitude towards both agents in terms of co-presence and building trust. These results challenge previous studies that favor human voice over TTS, and suggest that even if human voice is preferred, TTS can deliver equivalent benefits.

**Keywords:** embodied conversational agent; stress management; voice; text-to-speech; co-presence; trust; working alliance

## 1. Introduction

Intelligent virtual agents (IVAs) are artificially intelligent animated characters, designed to mimic natural human–human interaction for various objectives including entertainment [1], education [2], and healthcare [3]. Compared to conversational agents (like chatbots or voice assistants) that do not have a visual representation or embodiment, building acceptable and engaging IVAs for fruitful interactions is more challenging due to the appearance dimension included in the design of IVAs. For this reason, they are also commonly known as embodied conversational agents. The design of IVAs requires multidisciplinary effort, including psychology [4] and artificial intelligence [5] expertise to achieve capabilities such as emotion modeling [6,7]. Yet, these capabilities are largely still under laboratory investigation, and what factors may influence their acceptance by users is an ongoing research question [3,8].

People tend to perceive the presence, and consequently interact socially, with computers in similar ways as they naturally do with each other [9,10]. With the increasing use of IVAs in social, entertainment, health, and work settings, the acceptance of this technology is growing. Great effort is being devoted to increase this acceptance by giving the technology life by simulating the natural look and behavior of a human being (i.e., anthropomorphism) [11]. However, the theory of the "uncanny valley", which was first introduced by Mori [12], predicts that this acceptance will not last, and at some point, IVAs will manifest an unacceptable behavior/appearance and induce a feeling of eeriness and discomfort.

According to Nowak [13], presence has three measurable dimensions: social presence, co-presence, and presence as transportation. Social presence evaluates the ability of the media itself to bring the sense of presence/existence, co-presence evaluates if the sense of presence is established (mutual awareness and understanding), and presence as transportation evaluates the sense of being immersed in the environment as if the user has moved to the virtual world.

This differentiation between social presence and co-presence is important to understand the impact of anthropomorphism. In the literature, it is assumed that increasing the agent's realism (anthropomorphism) leads to higher social presence, and consequently increases the IVA's likeability and acceptance (e.g., [14,15]), with contradictory findings regarding the impact of anthropomorphism on the user–agent relationship and interaction goal (e.g., [16–19]). However, considering the uncanny valley problem, Brenton et al. [20] suggested that the eeriness problem or "uncanny valley" occurs as a result of how the user perceives the agent's co-presence. Co-presence is defined as having the sense of perceiving the IVA by the user and having the sense of being actively perceived by the IVA as well [13]. This mutual awareness is a conscious process that is developed when an interaction goal is established [13]. This conscious awareness and established goal help the user to adapt to a low level of features (low anthropomorphism) when, for example, using talking objects (agents) and interacting socially with them [21,22]. On the other hand, social presence is defined as having the sense of being 'there', in the same environment with the virtual agent. This clear distinction makes measurement of social presence more appropriate in virtual reality studies, such as [23]. Thus, in our study's context of conversing with an IVA, we focus on measuring the co-presence rather than the social presence.

The study of anthropomorphism has included the agent's appearance, facial expressions, voice, spoken responses, and physical movements. Due to the need for speech to be adaptive and the predominant use of synthetic voice/text-to-speech (TTS) technology rather than recorded real human voice, we focus our current research on the role of voice anthropomorphism (human vs. synthetic voice), its effect on co-presence, and its impact on the interaction outcome (intentions to change a behavior).

This study is part of a project where we investigate different factors of building a user–agent working alliance, including explanation and empathy [24–26]. Some of the participants in the previous experiments of the project complained about the voice being robotic. We sought to investigate the impact of this factor on our main desirable outcome (trust and behavior change), mainly to validate our previous findings as not being influenced by the voice type. The same concern applies to most of the current studies in the literature, which use TTS in building their chatbot or virtual agents. Our study contributes to the literature in validating others' findings.

In this paper, we argue that a virtual character with human voice will enhance the IVA's co-presence and consequently its likeability, and that the IVA's voice will impact the user–agent trust and interaction outcome. After a brief review of the literature in the following section, we describe our experiment, results, discussion, future work, and conclusions.

## 2. Related Work

### 2.1. Anthropomorphism and Co-Presence

People naturally enter into others' presence with very little information about the interaction situation, and hence they pay more attention to others' appearance and social cues [27]. The same rule applies when people interact with a human being or virtual character [9,28]. Presence has been considered as a major tool to evaluate social virtual agents by measuring to what extent the agent could engage the human user in the virtual world.

According to the ethopoeia theory, adding human social characteristics to an object increases the human's tendency to unconsciously accept the object as a social entity [9]. However, the threshold model of social influence posits that these characteristics should be as real as those of human beings (anthropomorphism) to boost a virtual agent's presence [29]. However, realism and presence are not always in a direct proportional rela-

tionship, and there is a point where eeriness is sensed—the uncanny valley problem [20] in human–agent interaction—which has still not been fully explained. Some researchers regard the problem as a subconscious cognitive reaction when the user's expectation is violated [30], while others regarded it as an emotional arousal response to the agent's appearance and behavior, correlated with the agent's presence [31]. Therefore, measuring the agent's presence, specifically co-presence, can reveal when the eeriness occurs as a result of anthropomorphism [20].

Voice is one of the human characteristics that increase the IVA's realism. It was found to be of great influence in applying social rules to objects where people perceived different voices of the same object as different agents [32]. A simple talking object, a tissue box saying "bless you", was perceived as social, as a humanoid robot, and as a human being [22]. Although the participants in [22] rated the robot and the box as eerier and less human than the human being, they rated them as being as social, attractive, friendly, and intelligent as the human being.

Voice anthropomorphism boosts the users' likeability [33]; however, there is a debate on its influence on the final interaction outcome. On one hand, some studies found anthropomorphism contributed towards increasing human–agent understanding, trust, connectedness, and task outcome [16,17,34,35]. On the other hand, other studies reported no influence on the outcome [36,37], or they had concerns with the use of higher anthropomorphism. For example, adding a human tone to the chatbot's voice influenced the user's online shopping behavior. A corporate's formal and emotionless tone, compared to an employee's informal and emotional tone, was perceived to be less credible when making important decisions during shopping, such as when shopping for gifts [38], and to have a higher privacy risk, which impacts the behavioral intention—to register on the business website [18].

Therefore, following the threshold model of social influence (higher realism leads to a greater sense of co-presence) and the assumption of Brenton, Gillies, Ballin, and Chatting [20] (measuring the perception of co-presence can explain when the voice eeriness occurs), we hypothesize the following:

**Hypothesis 1 (H1).** *The IVA with human voice will enhance the agent's co-presence compared to the IVA with TTS.*

**Hypothesis 2a (H2a).** *The IVA with higher co-presence will be rated with a higher likability.*

**Hypothesis 2b (H2b).** *The IVA with higher co-presence will be rated with a better voice impression.*

In a recent study by Higgins, Zibrek, Cabral, Egan, and McDonnell [23], they measured the social presence of a highly realistic IVA with TTS or real voice. They found that the mismatch, between realistic appearance and TTS, lowered the social presence of the IVA. In our study reported in this paper, we are also interested in measuring the impact of the mismatch, but with the use of artificial appearance and neutral voice or TTS. We are further measuring the co-presence rather than social presence for the abovementioned reason. The following section discusses the literature regarding the agent's anthropomorphism and congruence.

## 2.2. Anthropomorphism and Congruence

Early works in psychology concluded that humans naturally tend to prefer clarity and consistency in perception and interaction. The principle of grouping (Gestalt theory) postulates that people always perceive and organize surroundings/objects according to certain rules, including similarity [39]. When inconsistency occurs, the human brain gets confused and takes a longer time to respond, which is called the Stroop effect [40]. This confusion is not only limited to perceiving objects but extends to almost everything, including perceiving other humans and their personalities. The auditory Stroop effect was examined by Green and Barber [41], where their study participants found it more difficult

to identify the gender of a speaker when a male speaker says 'girl' or a female speaker says 'man'.

The same experience is extended to human–computer and human–robot interaction. Isbister and Nass [4] underlined the importance of the consistency in verbal and non-verbal cues of a virtual character, regardless of its personality or that of the users. Mitchell, Szerszen Sr, Lu, Schermerhorn, Scheutz, and MacDorman [36] reported that users found that a human face with a synthetic voice or a humanoid robot with a human voice caused significantly higher eeriness than when the face and voice were matched. Users rated the mismatched/inconsistent scenarios as eerier than the matched ones and the robot with a synthetic voice as the warmest. Similar results were reported earlier by Gong and Nass [42] using a human face or humanoid face matched with human or computerized voices. Moore [43] proved this phenomenon of the mismatch effect mathematically using a Bayesian model of categorical perception and called it 'perceptual tension'.

The impact of matching the agent's look with its voice (congruency) on user–agent trust has been further studied, with contrary findings. For example, Torre et al. [44] introduced an investment game to 120 participants to play with either a generous or a mean robot that switches from a human to a synthetic voice (or vice versa) in the middle of the game. The results indicate the importance of matching the voice with the look to form the first impression, which impacts the user's trust in the agent. However, other studies endorsed the use of a human recorded voice over a synthetic voice, such as Chérif and Lemoine [45], who concluded that a virtual assistant with a human voice on commercial websites increases the assistant's presence, and the users' behavioral intentions as well (N = 640), but it does not impact the users' trust in the website. They reported a negative influence of TTS on user–agent trust, which then negatively impacts behavioral intentions.

Further studies failed to find a significant difference between the use of human recorded voices and TTS regarding the system's desired outcome. With 138 undergraduate students, Zanbaka, Goolkasian, and Hodges [35] investigated how different persuasive messages would be perceived by students when the messages were delivered by human beings, IVAs, and a talking cat. The three settings had two versions: male and female. Human voices were used in the three settings. The results showed that participants perceived the IVAs and cats to be as persuasive as human speakers. Further, they showed a similar attitude towards the virtual characters and the human—male participants were more persuaded by real and virtual female speakers, while female participants were more persuaded by real and virtual male speakers. This stereotypical response is in line with the findings from voice-only studies by Mullennix, Stern, Wilson, and Dyson [33]. Similarly, other studies reported no significant differences between voice settings in terms of social reactions [46] or keeping a distance between a user and a robot [47].

The impact of the agent's voice type (human recorded voice vs. synthetic/computerized voice) could depend on context. In a learning scenario, Dickerson et al. [48] reported that participants found synthetic voice unnatural, but they quickly adapted and rated it as being as intelligent as the agent with a human voice. The system outcome was met equivalently in both groups, but they concluded that a human voice is preferred only for expressive virtual agents, as it can convey the speaker's emotions and attitudes. In a voice-only system, Noah et al. [49] concluded that synthetic voice would harm the user's trust in high-risk and highly personal contexts and that the use of human or synthetic voice should be investigated before deploying a system by studying the user's expectations and concerns with the type of the voice.

In a matching task, Torre, Latupeirissa, and McGinn [44] asked 60 participants to match one robot out of eight robots' pictures with a voice from a set of voices that varied in terms of gender and naturalness, for four different contexts: home, school, restaurant, and hospital. The results showed that the more mechanical the look of the robot, the greater the tendency for the robot to be matched with a synthetic voice, and that the matching was significantly different across the contexts.

Black and Lenzo [50] suggested the use of domain-limited (professional vs. general) synthetic voice to replace the use of human voice; however, Georgila et al. [51] reported no significant difference between a general-purpose synthetic voice and a good domain-limited synthetic voice in terms of perceived naturalness, conversational aspects, and likeability.

So far, we can conclude that the debate over the use of synthetic voice vs. human voice for IVAs is an ongoing problem, and that investigation has mainly been limited to the impact of the voice on the perception of the IVAs' likeability and acceptance, and less (with contradictory findings) on the human–agent relationship and interaction outcome [52]. In a recent comprehensive comparison between different types of voices, Cambre et al. [53] concluded that there is no one optimal voice type, and the choice depends on the context and the desired outcome of the system. Hence, we sought to contribute to the investigation of this problem in the domain of health behavior change, where the goal of the system is to increase the user–agent trust and the health behavior change intention. Thus, we expected the following:

**Hypothesis 3 (H3).** *There is a difference in the user–agent trust between those who interact with an IVA with human voice and those who interact with an IVA with TTS.*

**Hypothesis 4 (H4).** *There is a difference in the users' behavior change intention between those who interact with an IVA with human voice and who interact with an IVA with TTS.*

### 3. Methodology

*3.1. Study Design and Materials*

We conducted a between-subjects experiment approved by our university Human Ethics Committee in which the virtual agent has one of two voice settings: human voice or TTS voice. Due to potential carryover effects, we chose not to expose participants to both voice settings. First, for the appearance side of the agent, we designed the agent using Unity 3D as a female virtual advisor (VA), Figure 1, who interacts with the student to advise them on how to manage their study-related stress. The choice of a female VA was arbitrary, as the agent's gender has little to no influence on the user experience and participants tend to prefer younger female agents [54]. The advice was adapted from documents written by experts in the well-being clinic in the university. We converted these documents into a dialogue form (dialogue tree) to deliver the advice in a conversational manner between the agent and the student. In this study, the agent recommends the students to do four activities: join a study group, do physical activity, meet new people, and reduce caffeine consumption. We focus on these four behaviors because, in our previous studies [24,55], we found following them to be more challenging for the students. The design of the experiment in this study closely follows the design of the previous experiments in terms of agent and dialogue design, except they used TTS voice only.

Second, for the cognitive side of the agent, which controls its behavior, we utilized the cognitive agent architecture FAtiMA (FearNot affective mind architecture) [56]. More details on how we used and modified FAtiMA in our work can be found in [24]. The FAtiMA toolkit allows us to manage the agent–user dialogue by controlling the transition/turn-taking between the agent and the user according to the user's response. For example, as illustrated in Figure 1, based on the user's selection, the agent's response varies. Once the dialogue is scripted, the toolkit generates the audio files for all the agent's utterances using TTS with any voice of choice (e.g., the voices available in Microsoft windows language packs).

We generated the agent's utterances of the same dialogue in two ways to build two VAs: VA with human recorded voice (VA-recorded) and VA with TTS (VA-TTS). In the first setting, VA-recorded, the utterances were recorded by the voice of the second author, who is a female native English speaker. In the second setting, VA-TTS, the utterances were generated using Microsoft Hazel, the UK English voice.

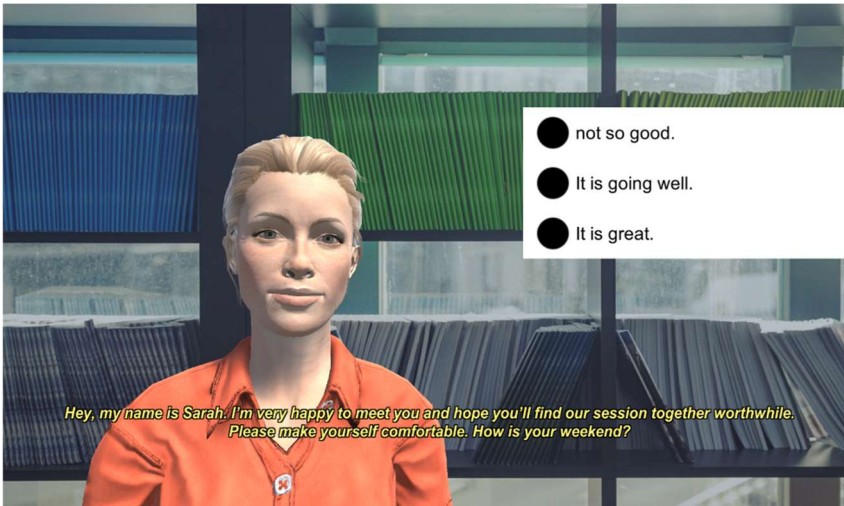

**Figure 1.** A snippet to illustrate the front-end design of the application. The figure shows the virtual advisor, Sarah, greeting the user and starting a small talk according to the time of the interaction (e.g., weekend or weekday). The user can select a response from a list of possible choices on the right of the screen.

Although synthetic voices with higher quality have been introduced [57,58], we opted to use the basic synthesized voice of Hazel for two reasons. First, some higher-quality synthesized voices could closely imitate the human voice (e.g., Sisman et al. [59] and Shen et al. [60]). Such a high-quality voice may be perceived as a human voice rather than a synthetic voice and impact the experiment objectives—studying the difference in the influence of synthetic vs. natural human voice. High-quality voices may confuse the user regarding whether it is a human or a machine voice, and this could be perceived at some point as a deception, which would directly harm the trust in the IVA [49]. We sought to be clear that this voice is a machine-generated voice without a warning, similar to the setting in [23]. Second, the VA is a downloadable application, and the use of available synthetic voices in the users' machines would avoid any technical problems due to the requirement of connecting to the servers to fetch the generated agent's voice.

*3.2. Recruitment, Instruments, and Procedure*

Participants were recruited through a university recruitment portal for undergraduate students. The study was advertised to students as a technology-mediated study to provide study stress tips. Typically, university students have access to the well-being center website, where they can receive a set of written tips to manage their study-related stress. We designed the dialogue of the VA to recommend those tips to the participants in a conversational style. In order to measure the influence of the VA on the behavior change intentions, we designed a behavior change intention questionnaire that includes the tips/behaviors recommended by the VA and measures the difference in the participants' intentions to do these behaviors before and after the interaction. In this study, the VA recommends four behaviors, which have been found to be more resistant change by the university students [24,55]. Further, to evaluate the suitability of using IVAs in the context of stress management for university students, we asked the students to report their study stress before and after the experiment on a scale of 0 to 10, where 0 means not being stressed at all and 10 means extremely stressed.

To evaluate the participants' impression of the VA, we used the likeability scale questions from the GodSpeed questionnaire introduced by Bartneck et al. [61]. The reliability test for the items using Cronbach $\alpha$ was 0.862. Ho and MacDorman [62] suggested adding another scale to evaluate the perception of the agent's voice independently from the perceived humanness of the agent, and they called it eeriness.

In order to measure the social relationship developed between a participant and the agent, we utilized two questionnaires: trust and working alliance inventory (WAI). The first instrument was developed by Mayer and Davis [63] and consists of five scales: trust propensity, three trustworthiness factors (ability, benevolence, and integrity), and trust. The first scale measures the participants' propensity to trust strangers in general. This scale provides a baseline to compare the experimental groups' attitudes towards trust in general. The trustworthiness scales include ability, which measures to what extent a participant believed in the agent's ability to do its job; benevolence, which measures the extent to which a participant believed the agent cares and wishes to do good to him/her; and integrity, which measures the extent to which the participants believed the agent followed ethical and moral principles. The final scale is trust, which includes items directly asking about the extent a participant trusted the agent. The items were adapted to suit the context (i.e., agent's name), and the reliability of the questionnaire was Cronbach $\alpha = 0.829$.

While the trust questionnaire could capture to what extent a participant would trust the VA for its capabilities, the WAI [64], particularly the bond scale, would capture the trust that results from the user's positive feeling of being mutually connected with the VA. We used the short form of WAI [65], which includes 12 items (Cronbach $\alpha = 0.936$), with 4 items to measure each of the following constructs: task, goal, and bond. Five-point Likert scales are used in the trust (strongly disagree—strongly agree) and working alliance (seldom—always) questionnaires. Furthermore, the participants have the option to select another option (not applicable) to give the participants the chance if he/she thinks the item is not appropriate or not applicable for VAs, due to the fact these questionnaires were originally designed to evaluate human–human interaction.

In order to measure if participants would perceive and sense the VA-TTS and the VA-recorded differently, we included the co-presence questionnaire by Nowak [13]. Cronbach $\alpha$ of this questionnaire was 0.731.

As illustrated in Figure 2, following consent, participants received a demographic questionnaire, propensity questions from the trustworthiness questionnaire, and behavior change intentions questionnaire. They were asked to score their emotional feelings towards their studies on a score of zero (extremely relaxed) to ten (extremely stressed) and to confirm that the computer sound was on, in order to listen and interact with the VA. Participants were randomly assigned to interact with VA-recorded or VA-TTS.

The average time to interact with the VA was eight minutes. After the interaction, the participants were asked to score their emotional feeling again on a zero-to-ten scale. They further received the same behavior change intentions questions, likeability and voice eeriness, social co-presence, trustworthiness and trust, and working alliance questionnaires. Participants took between 20–30 min to complete the entire study.

The questionnaires were delivered through the Qualtrics system. The participants did the study on their own devices operated by Windows or Mac operating systems only. The data were analyzed using IBM SPSS statistics 25.

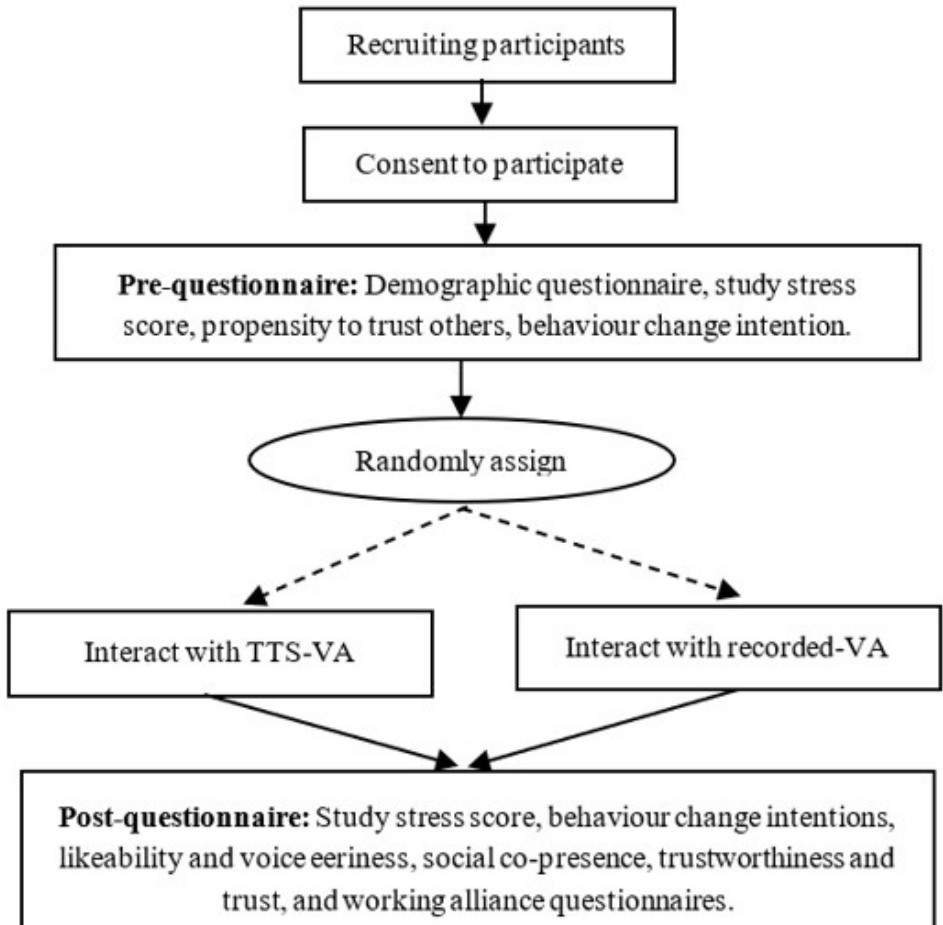

**Figure 2.** The study design flowchart.

## 4. Results

Out of 118 students who participated in the study, 97 of them were psychology students and 21 were from other fields. The participants' average age was 20.64 years old, with a standard deviation (STD) = 4.38. They were from different cultural groups, mainly the following: 29.3% Oceania, 19.9% South-east Asia, 13.8% Northern-western European, 7.7% Southern-Eastern European, and 6.6% North African and Middle Eastern combined. Table 1 presents the distribution of the participants in the two groups: the first group who interacted with the VA-TTS and the second group who interacted with the VA with recorded utterances by a human voice.

**Table 1.** Participants' distribution based on gender in the two experimental groups.

| Group | Female | Male | Total |
|:---:|:---:|:---:|:---:|
| TTS | 44 | 15 | 59 |
| Recorded | 36 | 23 | 59 |
| Total | 80 | 38 | 118 |

### 4.1. Study-Related Stress

Paired samples *t*-test revealed a statistically significant reduction in the study-related stress of the participants after interacting with both versions of the VA, as illustrated in Table 2. One-way ANOVA indicates no statistically significant between-groups difference (TTS vs. recorded) in terms of stress reduction ($F(1, 116) = 1.748$ at $p = 0.89$).

**Table 2.** Students' stress level statistics on a scale of 0 to 10 before and after the interaction.

| Group | Stress Level Before Interaction | | Stress Level After Interaction | | Paired Samples *t*-Test (Before vs. After) | |
|---|---|---|---|---|---|---|
| | **Mean** | **SDT** | **Mean** | **SDT** | **t** | **p** |
| TTS | 6.51 | 1.746 | 5.86 | 1.634 | 3.703 | **<0.001** |
| Recorded | 7.24 | 1.675 | 5.64 | 1.836 | 7.707 | <0.001 |

*4.2. Social Co-Presence*

Participants perceived the social co-presence of the two versions of the VA similarly. Table 3 presents the means and standard deviations of the six items delivered to the students after interacting with the VA. Independent-samples *t*-test found no statistically significant difference between the two groups in terms of the scores of co-presence items or their average (the co-presence as a construct) (at $p < 0.05$).

**Table 3.** Social co-presence items and statistics that were measured using five-point Likert scales (1: strongly disagree to 5: strongly agree).

| Items | Recorded | | TTS | |
|---|---|---|---|---|
| | **Mean** | **SDT** | **Mean** | **STD** |
| I did not want a deeper relationship with the agent * | 2.37 | 0.981 | 2.32 | 0.973 |
| I wanted to maintain a sense of distance between us * | 2.95 | 0.753 | 2.73 | 0.827 |
| I was unwilling to share personal information with the agent * | 3.31 | 1.178 | 3.36 | 0.905 |
| I wanted to make the conversation more intimate | 2.74 | 0.965 | 2.71 | 1.001 |
| I tried to create a sense of closeness between us | 2.66 | 0.843 | 2.64 | 0.846 |
| I was interested in talking to the agent | 3.22 | 0.966 | 3.25 | 1.03 |
| Average | 2.88 | 0.599 | 2.84 | 0.540 |

* Items are negatively stated, and the results are presented in the table after reversing the scores. The questionnaires used the name of the agent, but for blind review, we have replaced it with "the agent".

*4.3. Character Likeability and Voice Eeriness*

Independent-samples *t*-test reported no statistically significant difference between the likeability of IVA-recorded and IVA-TTS ($t(116) = 1.795$, $p = 0.08$). As presented in Figure 3, participants rated both versions of the VA as almost similar in terms of likability, where higher rates indicate a higher tendency towards the right pole of the question (i.e., like, friendly, kind, pleasant, nice). There was only a statistically significant between-group difference on the scale of awful—nice, favoring the VA-recorde ($t(116) = 2.335$ at $p = 0.021$).

Figure 4 presents the VAs' voice impression. Independent-samples *t*-test revealed statistically significant between-group differences in three items out of the eight items of the questionnaire. Compared to the VA-recorded, participants found VA-TTS eerier ($t(116) = -3.384$, $p = 0.001$), more supernatural ($t(116) = -2.854$, $p = 0.005$), and more boring ($t(116) = 2.270$, $p = 0.025$). However, *t*-test reported no statistically significant difference between the voice impression as a construct of IVA-recorded and IVA-TTS ($t(116) = -1.027$, $p = 0.31$).

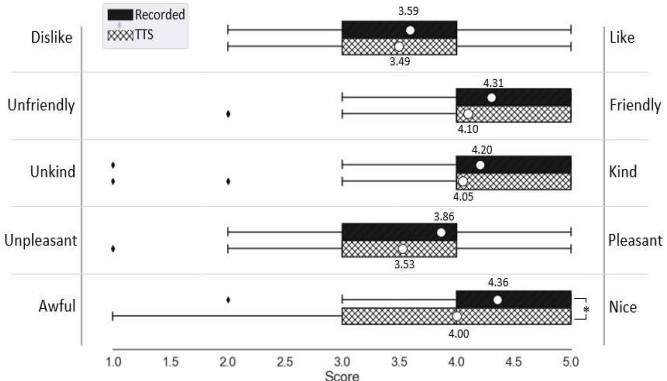

**Figure 3.** The VA likeability items statistics. The white circles mark the means.

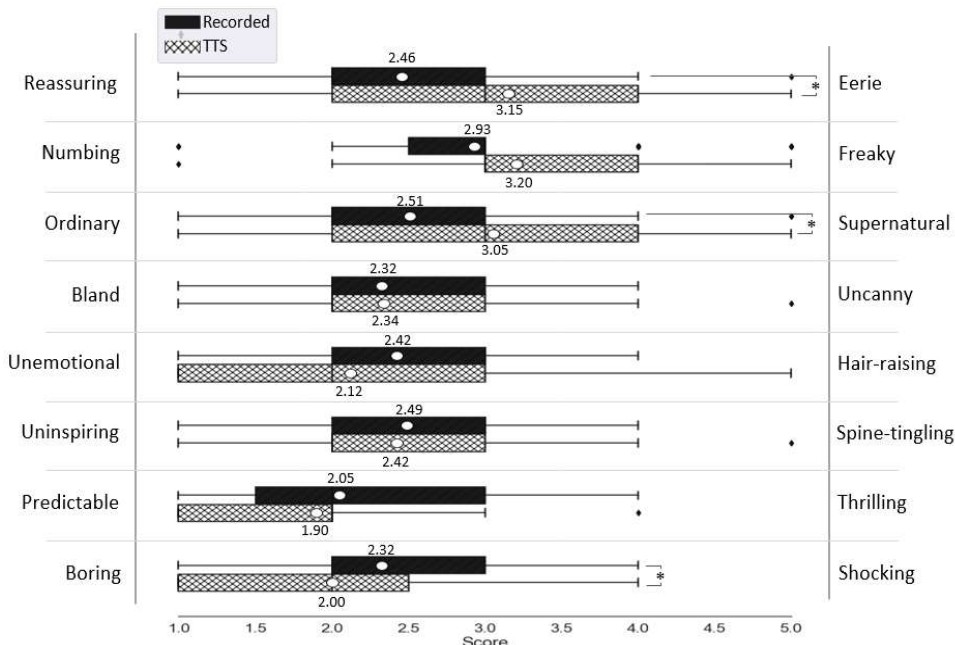

**Figure 4.** The VA voice eeriness items statistics. The white circles mark the means.

Regarding the understandability of the voice (Figure 5), participants evaluated the VA-recorded significantly easier to understand ($t(115) = 4.024$ at $p < 0.001$), and significantly easier to listen to its advice ($t(116) = 3.064$ at $p = 0.003$).

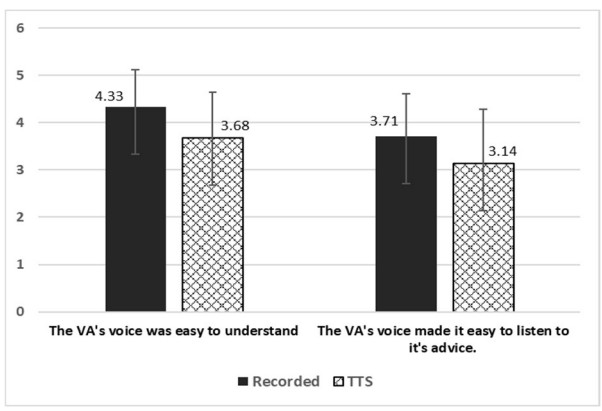

**Figure 5.** The VA's voice understandability.

Spearman's rank-order correlation was run to find the correlations between the co-presence and VAs' likeability and voice impression in both groups, VA-recorded and VA-TTS. Table 4 reports the correlation between the likeability and voice impression, as constructs and also their items, with co-presence. The likeability construct is calculated as the average of the five likeability items and the voice impression construct is calculated as the average of the eight voice impression items.

**Table 4.** VA's likeability and voice impression (as constructs and their items) correlations with co-presence.

| Measure | Recorded | | TTS | |
|---|---|---|---|---|
| | *r* | *p* | *r* | *p* |
| VA's likeability (the construct) | 0.380 | **0.003** | 0.511 | **<0.001** |
| Dislike—Like | 0.482 | **<0.001** | 0.486 | **<0.001** |
| Unfriendly—Friendly | 0.201 | 0.127 | 0.402 | **0.002** |
| Unkind—Kind | 0.058 | 0.663 | 0.258 | **0.048** |
| Unpleasant—Pleasant | 0.225 | 0.086 | 0.350 | **0.007** |
| Awful—Nice | 0.303 | **0.020** | 0.398 | **0.002** |
| VA's voice impression (the construct) | 0.176 | 0.181 | −0.301 | **0.020** |
| Reassuring—Eerie | −0.375 | **0.003** | −0.356 | **0.006** |
| Numbing—Freaky | −0.095 | 0.473 | −0.400 | **0.002** |
| Ordinary—Supernatural | −0.146 | 0.270 | −0.127 | 0.337 |
| Bland—Uncanny | 0.113 | 0.392 | 0.104 | 0.431 |
| Unemotional—Hair-raising | 0.042 | 0.751 | 0.025 | 0.849 |
| Uninspiring—Spine-tingling | 0.381 | **0.003** | −0.045 | 0.735 |
| Predictable—Thrilling | 0.367 | **0.004** | 0.103 | 0.437 |
| Boring—Shocking | 0.359 | **0.005** | 0.143 | 0.280 |

### 4.4. Trustworthiness, Trust, and Working Alliance

The statistics (means and STD) of the three scales of trustworthiness (ability, benevolence, and integrity), trust, and working alliance (task, goal, and bond) are reported in Table 5 for both experimental groups. There was a statistically significant difference only in terms of benevolence (between the two groups: $t(116) = 2.132$ at $p = 0.035$). Table 5 further reports how many times the participants selected the option "not applicable" for every construct. Every construct consists of a number of items. The column NA presents the sum of times the "not applicable" option was selected for all the items belonging to the specified construct. There was no significant correlation between co-presence and trustworthiness, trust, or working alliance.

### 4.5. Behavior Change Intention

Table 6 presents the means and standard deviations of the behaviors recommended by the VA to reduce study stress. Participants showed almost similar intention changes in both groups. The paired-samples $t$-test revealed that the change in intention to meet new people is not statistically significant in both groups. The intentions statistically significantly increased in both groups in terms of participating in study groups and statistically significantly decreased for consuming caffeinated food/drinks at $p < 0.05$. Regarding doing physical activity, the change in the intention was statistically significantly greater after interacting with the VA-TTS. However, the independent-samples $t$-test showed that the differences in the changes between the two groups for the four behaviors are not significant at $p < 0.05$.

**Table 5.** Trustworthiness, trust, and WAI results that were measured on five-point Likert scales. NA = not applicable.

| Construct | #Items | Recorded | | | TTS | | |
|---|---|---|---|---|---|---|---|
| | | NA | Mean | STD | NA | Mean | STD |
| | | | | Trustworthiness | | | |
| Ability | 5 | 0 (0%) | 3.30 | 0.794 | 0 (0%) | 3.28 | 0.622 |
| Benevolence * | 2 | 0 (0%) | 3.25 | 1.001 | 0 (0%) | 2.87 | 0.940 |
| Integrity | 2 | 0 (0%) | 3.72 | 0.665 | 0 (0%) | 3.82 | 0.700 |
| Trust | 4 | 0 (0%) | 2.95 | 0.884 | 4 (2%) | 2.85 | 0.682 |
| | | | | Working Alliance | | | |
| Task | 4 | 15 (6%) | 2.76 | 1.169 | 5 (2%) | 2.84 | 0.928 |
| Goal | 4 | 24 (10%) | 2.81 | 1.230 | 11 (5%) | 2.79 | 0.859 |
| Bond | 4 | 51 (22%) | 3.01 | 1.105 | 42 (18%) | 2.75 | 1.103 |

\* Significant between-group difference (recorded vs TTS) at $p < 0.05$.

**Table 6.** Behaviors recommended by the VA and the statistics that were measured using five-point Likert scale (1: never to 5: always).

| Behavior | VA-Recorded | | | | VA-TTS | | | |
|---|---|---|---|---|---|---|---|---|
| | Before Interaction | After Interaction | Paired Sample *t*-Test | | Before Interaction | After Interaction | Paired Sample *t*-Test | |
| | Mean (STD) | Mean (STD) | *t*(58) | *p* | Mean (STD) | Mean (SDT) | *t*(58) | *p* |
| Participate in a study group | 2.15 (1.00) | 2.42 (0.95) | −2.734 | **<0.01** | 2.05 (1.01) | 2.54 (0.97) | −3.959 | **<0.001** |
| Do physical activity | 2.34 (1.21) | 2.39 (1.05) | −1.524 | 0.133 | 2.31 (1.24) | 2.54 (1.15) | −2.188 | **0.033** |
| Meet new people | 2.66 (0.99) | 2.83 (0.95) | −0.369 | 0.713 | 2.68 (0.99) | 2.88 (1.00) | −1.651 | 0.104 |
| Consume caffeinated food/drink | 3.51 (1.15) | 3.00 (1.17) | 5.046 | **<0.001** | 3.54 (1.10) | 2.93 (1.17) | 5.788 | **<0.001** |

## 5. Discussion

The main aim of this study was to investigate the influence of voice anthropomorphism, human voice (recorded utterances) vs. machine-generated/synthetic voice (TTS), on the IVA's likeability, the human-agent trust, and the desired outcome of the interaction (behavior change intention). We followed the threshold model of social influence [29] in measuring the IVA's co-presence to understand the impact of the voice anthropomorphism on the factors of interest. We expected that the human voice would increase the perception of the IVA's co-presence; however, participants reported the same level of co-presence for the two versions of VAs (neutral on the Likert scale), and thus H1 was rejected. Although participants had the chance to select the questions of the co-presence as not applicable, they did not. This indicates the validity of the questions and that participants perceived the VAs as a social entity regardless of the voice type.

Participants rated VA-recorded slightly higher than VA-TTS in terms of likeability and more positively in terms of voice impression. They found TTS significantly more eerie than the recorded voice. Although there were some significant differences between the two types of VAs in terms of likeability and voice impression items, considering the likeability and voice impression as constructs, there was no between-group significant difference, leading us to reject H2a and H2b.

Table 4 reports that co-presence was positively significantly correlated with the likeability construct in both settings. Specifically, co-presence was significantly correlated with all the likeability items in the TTS group but only with two items in the recorded group.

This indicates that co-presence will be mostly correlated to the user's like or dislike of the voice (recorded or TTS) or, to a lesser extent, whether they find it awful/nice. However, participants' perceptions of the recorded voice for human-like qualities, such as friendly, kind, pleasant, were not significantly correlated to co-presence. For the TTS voice, however, we see participant's perception of these qualities correlated with their sense of co-presence. Although co-presence was only significantly correlated with the voice impression construct in the TTS group, both groups reported a significant negative correlation between voice eeriness and co-presence. Higher voice eeriness leads to less co-presence. This finding is in line with the threshold model: co-presence can predict the eeriness, when it occurs.

So far, the results of the VAs' likeability and voice impression do not provide evidence to support the claim in the literature that anthropomorphism through the use of human voice enhances the agent's likeability (e.g., Gong [66] and Chérif and Lemoine [45]). Brenton, Gillies, Ballin, and Chatting [20] assumed that the co-presence perception can capture eeriness, which could explain the agent-human relationship/likeability. The results of this study show that the VAs' likeability was explained to some extent by the voice eeriness, as there were moderately significant negative correlations between voice eeriness and likeability of VA-recorded ($r = -0.359$ at $p = 0.005$) and VA-TTS ($r = -0.477$ at $p < 0.001$).

The analysis failed to capture any between-group differences in terms of trustworthiness, trust, working alliance, and behavior change intentions as well, except in two cases: the benevolence scale in trustworthiness and meeting new people in the recommended behaviors. In the first case, benevolence, participants tended to rate the VA-recorded as more caring, but in the second case, meeting new people activity, participants were more persuaded by the VA-TTS to change their intentions to do this activity. Hence, with only minor between-group differences in the trustworthiness and behavior change intentions and weak correlations, we could not find any support for H3 or H4.

Whether anthropomorphism is required in designing IVAs/VAs is debatable, as explained in the literature review. Although we found that users would like and prefer an anthropomorphic voice to some extent, this likeability does not influence the user-agent relationship or interaction outcome. Therefore, we assume our findings align with the previous findings concluding that naturalness may not be important (e.g., [46,47]). Furthermore, considering the behavior change intention results presented in Table 6, we tend to agree with researchers who concluded that anthropomorphism may impair the interaction outcome. With the prevalence of graphical IVA, the inconsistency or mismatch in the agent's appearance and voice may distract the users from the original goal of the interaction. The failure in capturing a significant difference between human voice vs. synthetic voice could be deemed as an advantage for the use of synthetic voice, which facilitates building more adaptive conversational agents, where the dialogue is a collection of various information collected from the user during the interaction, such as user's name, user's favorite game, and current date.

Participants provided some insightful feedback on the VA's voice that can explain their attitude towards the VA-TTS. Some participants thought the VA was acceptable, though they would prefer a human voice: "The agent was great, maybe using a human voice over would have been more reassuring given what was being said". On the other hand, another participant commented, "It was robotic which was kind of off-putting and boring"; however, this particular participant rated his trust on the VA with 4/5, 3/5 on the VA's ability scale, 4/5 on the VA's integrity scale of trustworthiness, 3.75/5 on the task scale, 4/5 on the bond scale, and 3.50/5 on the goal scale of the working alliance. We can conclude that people would prefer to interact with conversational agents with more human-like voice, but the voice does not interfere with their social perception of the agent. Mullennix, Stern, Wilson, and Dyson [33] found that users prefer a conversational agent with TTS voice with matching gender. One male participant in our study who interacted with the VA-TTS stated "Make a male AI if the gender selected was male and vice versa for female. It will always be more creepy to talk to a digital incarnation of the opposite sex", which confirms the findings in [33]. However, in our analysis, between-gender statistical analysis

did not reveal any differences in terms of likeability, voice eeriness, working alliance, trustworthiness (except for benevolence), or trust. Nevertheless, future work could follow the participant's suggestion to match human-agent gender to measure any impacts.

## 6. Limitations, Future Work, and Conclusions

The key purpose of our study was to ensure that delivering a message using a TTS voice, rather than prerecording a human voice, would achieve similar outcomes in our domain of health and well-being. The items measured in our study were chosen due to our interest in behavior change intention and the development of a working alliance. If our message had differed by who speaks (man, woman, child, specific speaker), what is said (phonetic content), or environment (open space, room of different volume, the presence of interference), we would have needed to measure those. For the current study, we could have measured aspects such as how the natural or TTS voice spoke (pitch frequency, intonation, place of logical stress, duration of phonetic segments, and pauses) or listener factors (such as perceived familiarity with the topic and the listener's level of intelligence). However, our study was motivated to compare if there were differences in the outcomes delivered by the natural voice and the TTS voices that we had available to us. We are interested in studying the impact of the voice types as a medium to transfer the message rather than the vocalics (voice characteristics) [52]. Each study will have elements of the user that are deemed relevant for that particular study. This study captured demographic data and measured the user's stress and behavior intentions before and after the interaction.

Similar to this project, we have delivered a number of interventions in student populations to deal with study stress in Australia [67] and in India [68]. Other stress-related applications could include stress in office workers or to support remote workers that might feel social isolation. In other projects where we have used IVAs to allow patients and families to discuss recommended treatments for pediatric incontinence [69] and sleep disorders, we also captured working alliance and child's health status, parents' health literacy, and adherence behaviors. Other current applications include the use of an IVA to help stroke survivors manage their stroke recovery using the Take Charge after Stroke program [70] and an IVA to deliver emotional regulation strategies for insurance claimants.

The majority of current IVAs are built using TTS rather than spoken/recorded voice because TTS technology offers the advantage of uttering customized sentences, such as addressing the user by name and other adaptations. The literature provides evidence that agents with human voice are preferable to agents with TTS. However, little is known about whether such preferences impact the human-agent social relationship, for example regarding trust (and working alliance in the health domain). Although this study provides evidence that type of voice, human recorded voice vs. TTS, is not a major issue in building user's trust and working alliance with IVAs, it is of interest to investigate the reasons behind this finding. The reasons could include the perceived functionality and usefulness of the technology, which may mitigate the TTS eeriness and increase trust [71]. Further, this study finding is limited to one-session interaction rather than a longitudinal human-agent relationship. More investigations should be conducted to measure if the use of TTS may impact the users' engagement or their intentions to continue interacting with the conversational agents for a long term, with multiple sessions of interactions. Where differences are found due to voice, it will be necessary to identify what aspects of the voice influence these differences to evaluate differences in the speech signals, including sender, content, quality, context, and to whom the message is being sent. As improvements in TTS continue, differences in voice quality may reduce, with possible impacts from an uncanny valley perspective.

**Author Contributions:** Conceptualization, A.A. and D.R.; methodology, A.A. and D.R.; software, A.A.; validation, A.A. and D.R.; formal analysis, A.A.; investigation, A.A. and D.R.; resources, A.A. and D.R.; data curation, A.A.; writing—original draft preparation, A.A. and D.R.; writing—review and editing, A.A. and D.R.; visualization, A.A.; supervision, D.R.; project administration, D.R.; funding acquisition, D.R. All authors have read and agreed to the published version of the manuscript.

**Funding:** This research was funded by an International Macquarie University Research Training Program (iMQRTP) scholarship—No. 2015113.

**Institutional Review Board Statement:** This review did not involve the conduct of any studies with human participants. All research studies conducted by the authors that build on this review or which are cited in this review were conducted according to the guidelines of the Declaration of Helsinki, and approved by the Human Research Ethics Committee, Macquarie University, with reference number 52020535012841, dated 25 May 2019.

**Informed Consent Statement:** Not applicable.

**Data Availability Statement:** All of the data is contained within the article.

**Conflicts of Interest:** The authors declare no conflict of interest.

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
