# Peer review of "Is Natural Necessary? Human Voice versus Synthetic Voice for Intelligent Virtual Agents"

_mti, doi:10.3390/mti6070051_

Round 1

Reviewer 1 Report

 Is natural necessary? Human Voice versus Synthetic Voice For Intelligent Virtual Agents

Rewiev

                The approach presented in the article is methodologically incorrect, since the concept of "robot voice" is not defined. There are many text-to-speech systems that differ in intelligibility, naturalness, and cognitive load on speech perception. It is known that a formant or compilation synthesizer significantly reduces speech intelligibility in the presence of even small noises, and also creates difficulties in perceiving messages from several phrases, up to a complete loss of intelligibility. On the other hand, TTS based on neural networks have intelligibility and naturalness comparable to real speech. The authors are aware of this, but they preferred a different system, without indicating how important it was for them to use a speech signal with an explicit characteristic of the robot.

            In fact, the experimental set up described was supposed to answer the question of how much more convincing human speech is than synthetic speech. The speech signal contains information about who speaks (man, woman, child, specific speaker), what he says (phonetic content), how he speaks (pitch frequency, intonation, place of logical stress, duration of phonetic segments and pauses), in what environment (open space, room of different volume, the presence of interference). The characteristics of a speech message are also affected by the listener's perceived familiarity with the topic and the listener's level of intelligence.  Any assessment of the preference of human speech over synthetic speech will turn out to be contradictory and unreliable if these factors are ignored.

Reviewer 2 Report

-The paper presents the results of a study investigating the effect of voice type (human and synthetic) in Intelligent Virtual Agents (IVAs) in likeability, voice impression and interaction outcome.

-In line 160, I think there is an extra comma between the words "privacy" and "risky".

-I would suggest that you check how to refer to authors when referencing, for example, in the text you have "Torre, Latupeirrissa [45] asked 60 participants...", and I think it should be "Torre et al. [45] asked 60 participants..." because that paper has more than two authors.

-Be consistent when presenting results in the paper. In line 269 you present the value ".731" and previously in the papers you have included a zero before the decimal values.

-It is interesting that according to the results obtained, participants perceived VAs as a social entity regardless of the voice type.

-There are a few grammatical errors and typos in the text. I recommend the authors to check the text carefully to amend them.

-Have you considered other application areas in which you could use your virtual advisor? For example, stress in office workers or to support remote workers that might feel social isolation?

-The technical aspects of the research presented seem sound and detailed.

-The results obtained seem to be promising for the authors' future work.

-The topic and area of research are relevant.

Reviewer 3 Report

Very interesting study. I have not many comments regarding this. Please update the state of art (related work - section) with same papers from 2021 or even 2022.

There is lack of further research plans. How do you plan to expand this work.

Reviewer 4 Report

The paper describes a study to decide whether TTS voices or human voices are better for IVAs to facilitate so called co-presence.

I have several concerns with this study:

a) The introduction is quite broad and a lot of concepts and terms are used without a further explanation. For example:

-That the authors are working with an avatar-based system instead of an interactive voice assistant (that maybe do not have a visual presence) is just knowable from later sections.

-why should IVAs mimic natural human-human interactions?

-what are believabe IVAs?

-what is the eeriness problem (not introduced in the introduction)

-The statement Co-presence is defined as having... (line 55ff) is mising a reference.

line 123: In this study -> in the actual study

The methods chapter leave the reader alone with a lot of design decisions the authors made:

a) why is just a between subjects design used and not a within-subject cross experiment (when the dialogs are just 8 minutes and some time used for the questionnairs (how much, never mentioned) there should be enough time for two rounds (random order)

b) Why do you use a female voice, does that have an influence on the perception?

c) You only mention the TTS voice but not the voice SDK/API or system.

d) What is about the relevance of your study, if you admit that there are better TTS-voices that are indistinguishable from real human voices. Wouldn't that mean that in a few years the TTS-quality does not have an influence on the co-presence at all? So why bother?

e) line 238 -> "Sara is downloadable" -> Who/What is Sara

f) regarding the conversation, a sample dialog would be helpfull

g) how can you guarantee that the pre-recorded utterances fit to the dialog?

h) The description of all the questionnaires is a bit confusing, a procedure scheme would be helpful.

Comments regarding the results section:

a) line 301 "6.6% North American and Middle East" each or both together?

b) Table 1: Why are the genders so imbalanced between the two conditions?

c) in section 4.3. the authors should use the same terms as used in the figures, especially "boring (line 345) is not visible in Fig2!.

In general, the authors should re-think the chosen presentation forms for the results. Sometimes, they used tables, sometime line-plots, and sometimes bar plots for the same type of data. Why?

This hinders a better comparison.

Furthermore, where is Fig 1 and why does Fig. 2 consists of two graphics with different aspect ratio, and the upper graphic is missing an x-axis.

Discussion:

In the discussion, the authors tried to refer their results back to their research hypotheses. But it could be depicted in a better way, as actually the different hypotheses are just mentioned in the text, without rephrasing the hypothesis text. Furthermore, I would recommend discussing the actual results in comparison to other results especially regarding [22], because it is a bit strange to start with the hypothesis that TTS and human voices have an influence but then all the assumptions fail.

Furthermore, I only noticed after reading the discussion that the participants gave verbal feedback.

Author Response

Open Review

(x) I would not like to sign my review report

( ) I would like to sign my review report

English language and style

( ) Extensive editing of English language and style required
( ) Moderate English changes required

(x) English language and style are fine/minor spell check required

( ) I don't feel qualified to judge about the English language and style

Yes

Can be improved

Must be improved

Not applicable

Does the introduction provide sufficient background and include all relevant references?

( )

( )

(x)

( )

Are all the cited references relevant to the research?

(x)

( )

( )

( )

Is the research design appropriate?

( )

( )

(x)

( )

Are the methods adequately described?

( )

( )

(x)

( )

Are the results clearly presented?

( )

(x)

( )

( )

Are the conclusions supported by the results?

( )

( )

(x)

( )

Comments and Suggestions for Authors

The paper describes a study to decide whether TTS voices or human voices are better for IVAs to facilitate so called co-presence.

I have several concerns with this study:

  1. a) The introduction is quite broad and a lot of concepts and terms are used without a further explanation. For example:

-That the authors are working with an avatar-based system instead of an interactive voice assistant (that maybe do not have a visual presence) is just knowable from later sections.

RESPONSE: We commenced our article with a definition of the term we use throughout, including in the title, to refer to the type of technology we are using: intelligent virtual agents. We are not using a voice assistant and we do not use that term anywhere. We are also not using an avatar-based system. The word avatar is not used once in our article. We are using an agent-based system, hence the use of the term “agent” and also the use of the term “intelligent”. We have not created this term. This term is widely known and used to refer to the AI-driven agents with visual appearance.

When we use the term agent, as used by IVA researchers, we refer to a computer-based agent that is artificially intelligent, as indicated by the term “intelligent”. To make this more apparent to readers not familiar with IVA technology or research in this field, we have added the words “artificially intelligent” to the definition in our opening sentence. Our description “animated characters, designed to mimic the natural human-human interaction” sought to convey it has a visual representation/embodiment. The other term commonly used for this technology is Embodied Conversational Agents, but we use the more general term used by the community. We chose not to introduce any further terms to avoid confusion. We could add the word “embodied” before “animated character”, but that seems to be redundant.

-why should IVAs mimic natural human-human interactions?

RESPONSE: Mimicking human-human interaction or “anthropomorphism” is an ongoing research area. The literature is rich with findings on how more humanlike features (e.g., look, sound, body gesture, behaviour, etc) embedded in IVAs could improve the user experience interacting with the IVA and could reap better system outcomes. For more information, we have cited a recent review ([15] in the paper) which reviewed the literature of anthropomorphism in the field of IVA.

Because the anthropomorphism topic is very broad, in our paper, we only focused on the findings of anthropomorphism and the related terms in the paper, for examples, social presence, likeability, and congruence.

-what are believable IVAs?

RESPONSE: the notion of “believable agent” is widely used in the field. The simplest definition is “..provides the illusion of life..” by Joseph Bates in “The role of emotions in believable agents”. IVA researchers use the term ‘believable’ to describe the agent’s human-like features such as believable emotion, behaviour, personality, and so on. Please note that, in our paper, we didn’t use this definition and we did not seek to measure believability. To avoid any confusion and decrease the definitions, we replaced the word ‘believable’ with ‘acceptable and engaging’.

-what is the eeriness problem (not introduced in the introduction)

RESPONSE: Eeriness problem is an alternative term to “the uncanny valley”. We added the term “uncanny valley” after the term “eeriness problem” in the paper:

However, considering the uncanny valley problem, Brenton, et al. [20] suggested that the eeriness problem or “uncanny valley” occurs as a result of how the user perceived the agent’s co-presence.”

-The statement Co-presence is defined as having... (line 55ff) is missing a reference.

RESPONSE: We added the reference.

line 123: In this study -> in the actual study

RESPONSE: We changed “In this study” to “In our study reported this paper,..” to distinguish between “the recent study” mentioned in the previous sentence and “our study”.

The methods chapter leave the reader alone with a lot of design decisions the authors made:

  1. why is just a between subjects design used and not a within-subject cross experiment (when the dialogs are just 8 minutes and some time used for the questionnairs (how much, never mentioned) there should be enough time for two rounds (random order)

RESPONSE: In line 296, we reported that “Participants were randomly assigned to interact with VA-recorded or the VA-TTS.” Hence, we claimed that our experiment is of between-subjects design. Participants did not interact with both virtual agents.

We reported an average of 8 minutes, but some participants took longer. We have up to 30 minutes for the study with all of the pre and post questionnaires. We have added the sentence “Participants took between 20-30 minutes to complete the entire study.” In line 305, modified version of the paper.

In addition to not having time for both a within and between subject design, we did not want to contaminate/bias the second interaction due to possible carryover of impressions from the first round. This design kept our statistical analyses simpler.

  1. Why do you use a female voice, does that have an influence on the perception?

RESPONSE: We have added the following statement to Section 3.1. Study Design and Materials.

The choice of a female VA was arbitrary as the agent’s gender has little to no influence on the user experience and participants tend to prefer younger female agent [51].

Furthermore, both the human and agent voice are female. So if there are any gender-related effect it should apply to both treatments.

In section 5, we discuss the character’s gender as part of responding to a participant’s comment. We also clarify that

 “However, in our analysis, between-gender statistical analysis did not reveal any differences in terms of likeability, voice eeriness, working alliance, trustworthiness (except for benevolence), or trust.”

We have now added to the end of this sentence “Nevertheless, future work could follow the participant’s suggestion to match human-agent gender to measure any impacts.”.

  1. You only mention the TTS voice but not the voice SDK/API or system.

RESPONSE: We did not need to use SDK/API. Our system (using FAtiMA) uses either generated voices (TTS technology) or recorded voices. We have clarified this point in the paper as follows:

“In the second setting, VA-TTS, the utterances were generated using UK English Hazel. Although synthetic voices with higher quality have been introduced [54,55], we opted to use the basic synthesised voice of Hazel for two reasons. First, some higher quality synthesised voices could closely imitate the human voice (e.g. Sisman, et al. [56] and Shen, et al. [57])….”

  1. What is about the relevance of your study, if you admit that there are better TTS-voices that are indistinguishable from real human voices. Wouldn't that mean that in a few years the TTS-quality does not have an influence on the co-presence at all? So why bother?

RESPONSE: This study is part of a project where we investigate different factors on building user-agent therapeutic alliance including explanation and empathy. Some of the participants in previous experiments of the project complained about the voice being robotic. We sought to investigate the impact of this factor on our main desirable outcome (trust and behaviour change), mainly, to validate our results of the project in general. We mean by validation here is to find if the type of voice is a cause of reduction in user-agent trust/alliance or in behaviour change. If there was a significant difference in our study, reported in this paper, favouring the natural/recorded voice, we should reconsider our design and previous results as well.

The same concern applies to most of the current studies in the literature that use TTS in building their chatbot or virtual agents. Our study contributes to the literature in validating others’ findings. Currently, the use of high-quality TTS is expensive to some extent or not available to some researchers.

  1. line 238 -> "Sara is downloadable" -> Who/What is Sara

RESPONSE: Sorry for this mistake. Sarah is the name of the VA in the experiment which we didn’t report it in the paper. We replaced the name “Sarah” with “the VA” in the paper.

  1. regarding the conversation, a sample dialog would be helpful

RESPONSE: We have published the dialogue in our previous papers which we will refer to at the publication stage. We didn’t present parts of the dialogue in this paper as our main focus is the voice type regardless the content, especially that the content of both VAs was identical.

  1. how can you guarantee that the pre-recorded utterances fit to the dialog?

RESPONSE: The dialogue is first crafted as text. Next, the text that represent the agent’s responses: 1) is used as input to the TTS generator and 2) is read by the female reader and recorded with the help of an expert using Audacity program. Every response of the agent is recorded as a separated audio file. Further, the text/response and its corresponding audio are paired programmatically for later use. In the dialogue tree (agent-user turn taking), the audio files (TTS or recorded) are used to generate the VA’s behaviour (speech): synchronizing the audio file with the VA’s lips’ movement, and their paired text are displayed on the screen as subtitles.

We can add this further description to the paper if deemed important for the reader.

  1. The description of all the questionnaires is a bit confusing, a procedure scheme would be helpful.

RESPONSE: A new figure (Figure 1) is added to clarify the study design flow chart.

Comments regarding the results section:

  1. line 301 "6.6% North American and Middle East" each or both together?

RESPONSE: yes, combined. We added the word at the end of the sentence “combined” after "6.6% North American and Middle East". Please note that the culture group question is designed to include the twelve main cultures in Australia, which are defined by the Australian Bureau of Statistics (2016).

  1. Table 1: Why are the genders so imbalanced between the two conditions?

RESPONSE: In every group, the female participants number is greater than the male. However, there was no significant difference between the two groups in terms of female to male distribution. This ingroup imbalance shouldn’t impact the results as we further ran different statistical tests to investigate if the gender (within the experimental groups) has an influence on other factors as reported at the very end of the discussion section:

“However, in our analysis, between-gender statistical analysis did not reveal any differences in terms of likeability, voice eeriness, working alliance, trustworthiness (except for benevolence), or trust.”

  1. in section 4.3. the authors should use the same terms as used in the figures, especially "boring (line 345) is not visible in Fig2!.

RESPONSE: With the new Figure 1, Figure 2 is now Figure 3. We can see the word “boring” at the bottom of Figure 3. We believe some error must have occurred in the creation of the pdf available to the reviewer.

In general, the authors should re-think the chosen presentation forms for the results. Sometimes, they used tables, sometime line-plots, and sometimes bar plots for the same type of data. Why?

This hinders a better comparison.

RESPONSE: Sorry if our choice of presentation of results caused any confusion. We tried to simplify the results for the reader by choosing the best presentation for every type of data. While we could report all the results in tables, we used visual presentation for likeability and voice impressions to show the tendency of the participants in each group towards the two poles. However, the voice understandability data was not applicable to be presented in a similar way, so we opted to use bar graph, again to visualize the results and reduce stuffing the paper with more numbers and tables.

Furthermore, where is Fig 1 and why does Fig. 2 consists of two graphics with different aspect ratio, and the upper graphic is missing an x-axis.

RESPONSE: Sorry, the graphs overlapped at some point before the submission, although we ensured everything was in place. We fixed the problem now.

Discussion:

In the discussion, the authors tried to refer their results back to their research hypotheses. But it could be depicted in a better way, as actually the different hypotheses are just mentioned in the text, without rephrasing the hypothesis text. Furthermore, I would recommend discussing the actual results in comparison to other results especially regarding [22], because it is a bit strange to start with the hypothesis that TTS and human voices have an influence but then all the assumptions fail.

RESPONSE: Thanks for your suggestion.

We followed a very standard and common method of investigating the impact of an independent variable (i.e., voice type) on dependent variables (co-presence, trust, likeability,…) with hypotheses formulation based on the literature. Hence, the discussion should clearly report how the results support or do not support these hypotheses (accept or reject the hypotheses). Failing to support the hypotheses means that the alternative hypotheses are true, and consequently, there is an alternative path for the researchers/readers to follow.

The study in [22] differs from our study so we sought not to compare our results to their results. The study in [22] used virtual reality environment rather than an intelligent virtual agent that is presented on any screen. They further measure social presence rather than co-presence. However, we discussed our results against the literature in the discussion (e.g., paragraphs start at line 424 and line 444 in the modified version of the paper)

Furthermore, I only noticed after reading the discussion that the participants gave verbal feedback.

RESPONSE: The participants provided feedback at the end of the study as a reply to a free-text question about their opinion on the study in general. There was no verbal feedback collected.

Round 2

Reviewer 4 Report

Dear authors, thank you for your extensive answers to my remarks. I tried to answer all of them below (bold text for the initial remarks and my answers to your answers).

Three minor remarks I still have:

a) present your implementation if the IVA much earlier in the motivation and add the motivation about the relevance of your study

b) give some more technical details (e.g. FAtiMA and the system you used to generate the TTS-voice.

c) justify why mimikry is important for IVAs

Otherwise, the actual manuscript is acceptable

The paper describes a study to decide whether TTS voices or human voices are better for IVAs to facilitate so called co-presence.

I have several concerns with this study:

  1. a) The introduction is quite broad and a lot of concepts and terms are used without a further explanation. For example:

-That the authors are working with an avatar-based system instead of an interactive voice assistant (that maybe do not have a visual presence) is just knowable from later sections.

RESPONSE: We commenced our article with a definition of the term we use throughout, including in the title, to refer to the type of technology we are using: intelligent virtual agents. We are not using a voice assistant and we do not use that term anywhere. We are also not using an avatar-based system. The word avatar is not used once in our article. We are using an agent-based system, hence the use of the term “agent” and also the use of the term “intelligent”. We have not created this term. This term is widely known and used to refer to the AI-driven agents with visual appearance.

When we use the term agent, as used by IVA researchers, we refer to a computer-based agent that is artificially intelligent, as indicated by the term “intelligent”. To make this more apparent to readers not familiar with IVA technology or research in this field, we have added the words “artificially intelligent” to the definition in our opening sentence. Our description “animated characters, designed to mimic the natural human-human interaction” sought to convey it has a visual representation/embodiment. The other term commonly used for this technology is Embodied Conversational Agents, but we use the more general term used by the community. We chose not to introduce any further terms to avoid confusion. We could add the word “embodied” before “animated character”, but that seems to be redundant.

I know that the term IVA is widely known and used, but there are different implementations. Researchers comprise voice assistants (i.e. smart speakers) as well as chatbots and agents having a virtual (human-like) representation (i.e. Max from uni bielefeld) as a IVA. Therefore, the term ECA (embodied conversational Agent) is used to specify a sub-group of IVAs.

Therefore, and to understand the experimental setting, this should be made clear in the beginning of your manuscript. Thus, I recommend to define your implementation of an IVA in the beginning if the introduction. Furthermore, a scree-shot of the agent would be good in the Methods chapter

-why should IVAs mimic natural human-human interactions?

RESPONSE: Mimicking human-human interaction or “anthropomorphism” is an ongoing research area. The literature is rich with findings on how more humanlike features (e.g., look, sound, body gesture, behaviour, etc) embedded in IVAs could improve the user experience interacting with the IVA and could reap better system outcomes. For more information, we have cited a recent review ([15] in the paper) which reviewed the literature of anthropomorphism in the field of IVA.

Because the anthropomorphism topic is very broad, in our paper, we only focused on the findings of anthropomorphism and the related terms in the paper, for examples, social presence, likeability, and congruence.

I’m not asking for a definitionh of mimikry, I was asking to give a justification why a ECA should do that, what are the assumptions regarding the dialog? Especially how are enhancing co-presence and likability/voice-impression related to mimikry?

-what are believable IVAs?

RESPONSE: the notion of “believable agent” is widely used in the field. The simplest definition is “..provides the illusion of life..” by Joseph Bates in “The role of emotions in believable agents”. IVA researchers use the term ‘believable’ to describe the agent’s human-like features such as believable emotion, behaviour, personality, and so on. Please note that, in our paper, we didn’t use this definition and we did not seek to measure believability. To avoid any confusion and decrease the definitions, we replaced the word ‘believable’ with ‘acceptable and engaging’.

Thanks!

-what is the eeriness problem (not introduced in the introduction)

RESPONSE: Eeriness problem is an alternative term to “the uncanny valley”. We added the term “uncanny valley” after the term “eeriness problem” in the paper:

However, considering the uncanny valley problem, Brenton, et al. [20] suggested that the eeriness problem or “uncanny valley” occurs as a result of how the user perceived the agent’s co-presence.”

Thanks!

-The statement Co-presence is defined as having... (line 55ff) is missing a reference.

RESPONSE: We added the reference.

Thanks!

line 123: In this study -> in the actual study

RESPONSE: We changed “In this study” to “In our study reported this paper,..” to distinguish between “the recent study” mentioned in the previous sentence and “our study”.

OK.

The methods chapter leave the reader alone with a lot of design decisions the authors made:

  1. why is just a between subjects design used and not a within-subject cross experiment (when the dialogs are just 8 minutes and some time used for the questionnairs (how much, never mentioned) there should be enough time for two rounds (random order)

RESPONSE: In line 296, we reported that “Participants were randomly assigned to interact with VA-recorded or the VA-TTS.” Hence, we claimed that our experiment is of between-subjects design. Participants did not interact with both virtual agents.

We reported an average of 8 minutes, but some participants took longer. We have up to 30 minutes for the study with all of the pre and post questionnaires. We have added the sentence “Participants took between 20-30 minutes to complete the entire study.” In line 305, modified version of the paper.

In addition to not having time for both a within and between subject design, we did not want to contaminate/bias the second interaction due to possible carryover of impressions from the first round. This design kept our statistical analyses simpler.

I’m not fully convinced by that, because possible carryover impressions could be avoided by having a random order of the experiments for all participants. But I know that it is hard to change a once conducted experiment, so maybe just add the aim to avoid carryover effects.

  1. Why do you use a female voice, does that have an influence on the perception?

RESPONSE: We have added the following statement to Section 3.1. Study Design and Materials.

The choice of a female VA was arbitrary as the agent’s gender has little to no influence on the user experience and participants tend to prefer younger female agent [51].

Furthermore, both the human and agent voice are female. So if there are any gender-related effect it should apply to both treatments.

In section 5, we discuss the character’s gender as part of responding to a participant’s comment. We also clarify that

However, in our analysis, between-gender statistical analysis did not reveal any differences in terms of likeability, voice eeriness, working alliance, trustworthiness (except for benevolence), or trust.”

We have now added to the end of this sentence “Nevertheless, future work could follow the participant’s suggestion to match human-agent gender to measure any impacts.”.

Thanks, that what I intended, to have an additional discussion of that point

  1. You only mention the TTS voice but not the voice SDK/API or system.

RESPONSE: We did not need to use SDK/API. Our system (using FAtiMA) uses either generated voices (TTS technology) or recorded voices. We have clarified this point in the paper as follows:

“In the second setting, VA-TTS, the utterances were generated using UK English Hazel. Although synthetic voices with higher quality have been introduced [54,55], we opted to use the basic synthesised voice of Hazel for two reasons. First, some higher quality synthesised voices could closely imitate the human voice (e.g. Sisman, et al. [56] and Shen, et al. [57])….”

I think to understand the experiment and the utilized technique (recapitulation and comparisum of experiments), it is important to get the information of the framework (FatiMA) and the TTS-voice. And I still do not get the point about the voice. You wrote you use UK English Hazel, but from which system: MaryTTS, Amazon Polly, etc? This is important, if others do similar experiments with other voices to then compare the voice impression also acoustically.

  1. What is about the relevance of your study, if you admit that there are better TTS-voices that are indistinguishable from real human voices. Wouldn't that mean that in a few years the TTS-quality does not have an influence on the co-presence at all? So why bother?

RESPONSE: This study is part of a project where we investigate different factors on building user-agent therapeutic alliance including explanation and empathy. Some of the participants in previous experiments of the project complained about the voice being robotic. We sought to investigate the impact of this factor on our main desirable outcome (trust and behaviour change), mainly, to validate our results of the project in general. We mean by validation here is to find if the type of voice is a cause of reduction in user-agent trust/alliance or in behaviour change. If there was a significant difference in our study, reported in this paper, favouring the natural/recorded voice, we should reconsider our design and previous results as well.

The same concern applies to most of the current studies in the literature that use TTS in building their chatbot or virtual agents. Our study contributes to the literature in validating others’ findings. Currently, the use of high-quality TTS is expensive to some extent or not available to some researchers.

OK, can you put that soimehow into the motivation?

  1. line 238 -> "Sara is downloadable" -> Who/What is Sara

RESPONSE: Sorry for this mistake. Sarah is the name of the VA in the experiment which we didn’t report it in the paper. We replaced the name “Sarah” with “the VA” in the paper.

OK.

  1. regarding the conversation, a sample dialog would be helpful

RESPONSE: We have published the dialogue in our previous papers which we will refer to at the publication stage. We didn’t present parts of the dialogue in this paper as our main focus is the voice type regardless the content, especially that the content of both VAs was identical.

OK.

  1. how can you guarantee that the pre-recorded utterances fit to the dialog?

RESPONSE: The dialogue is first crafted as text. Next, the text that represent the agent’s responses: 1) is used as input to the TTS generator and 2) is read by the female reader and recorded with the help of an expert using Audacity program. Every response of the agent is recorded as a separated audio file. Further, the text/response and its corresponding audio are paired programmatically for later use. In the dialogue tree (agent-user turn taking), the audio files (TTS or recorded) are used to generate the VA’s behaviour (speech): synchronizing the audio file with the VA’s lips’ movement, and their paired text are displayed on the screen as subtitles.

We can add this further description to the paper if deemed important for the reader.

Oh sorry for the confusion, I meant if the participants can answer freely, how can you guarantee that you have a suitable pre-recorded answer?

  1. The description of all the questionnaires is a bit confusing, a procedure scheme would be helpful.

RESPONSE: A new figure (Figure 1) is added to clarify the study design flow chart.

Thanks!

Comments regarding the results section:

  1. line 301 "6.6% North American and Middle East" each or both together?

RESPONSE: yes, combined. We added the word at the end of the sentence “combined” after "6.6% North American and Middle East". Please note that the culture group question is designed to include the twelve main cultures in Australia, which are defined by the Australian Bureau of Statistics (2016).

Thanks

  1. Table 1: Why are the genders so imbalanced between the two conditions?

RESPONSE: In every group, the female participants number is greater than the male. However, there was no significant difference between the two groups in terms of female to male distribution. This ingroup imbalance shouldn’t impact the results as we further ran different statistical tests to investigate if the gender (within the experimental groups) has an influence on other factors as reported at the very end of the discussion section:

“However, in our analysis, between-gender statistical analysis did not reveal any differences in terms of likeability, voice eeriness, working alliance, trustworthiness (except for benevolence), or trust.”

OK Thanks!

  1. in section 4.3. the authors should use the same terms as used in the figures, especially "boring (line 345) is not visible in Fig2!.

RESPONSE: With the new Figure 1, Figure 2 is now Figure 3. We can see the word “boring” at the bottom of Figure 3. We believe some error must have occurred in the creation of the pdf available to the reviewer.

Thanks

In general, the authors should re-think the chosen presentation forms for the results. Sometimes, they used tables, sometime line-plots, and sometimes bar plots for the same type of data. Why?

This hinders a better comparison.

RESPONSE: Sorry if our choice of presentation of results caused any confusion. We tried to simplify the results for the reader by choosing the best presentation for every type of data. While we could report all the results in tables, we used visual presentation for likeability and voice impressions to show the tendency of the participants in each group towards the two poles. However, the voice understandability data was not applicable to be presented in a similar way, so we opted to use bar graph, again to visualize the results and reduce stuffing the paper with more numbers and tables.

OK!

Furthermore, where is Fig 1 and why does Fig. 2 consists of two graphics with different aspect ratio, and the upper graphic is missing an x-axis.

RESPONSE: Sorry, the graphs overlapped at some point before the submission, although we ensured everything was in place. We fixed the problem now.

Thanks!

Discussion:

In the discussion, the authors tried to refer their results back to their research hypotheses. But it could be depicted in a better way, as actually the different hypotheses are just mentioned in the text, without rephrasing the hypothesis text. Furthermore, I would recommend discussing the actual results in comparison to other results especially regarding [22], because it is a bit strange to start with the hypothesis that TTS and human voices have an influence but then all the assumptions fail.

RESPONSE: Thanks for your suggestion.

We followed a very standard and common method of investigating the impact of an independent variable (i.e., voice type) on dependent variables (co-presence, trust, likeability,…) with hypotheses formulation based on the literature. Hence, the discussion should clearly report how the results support or do not support these hypotheses (accept or reject the hypotheses). Failing to support the hypotheses means that the alternative hypotheses are true, and consequently, there is an alternative path for the researchers/readers to follow.

The study in [22] differs from our study so we sought not to compare our results to their results. The study in [22] used virtual reality environment rather than an intelligent virtual agent that is presented on any screen. They further measure social presence rather than co-presence. However, we discussed our results against the literature in the discussion (e.g., paragraphs start at line 424 and line 444 in the modified version of the paper)

Furthermore, I only noticed after reading the discussion that the participants gave verbal feedback.

RESPONSE: The participants provided feedback at the end of the study as a reply to a free-text question about their opinion on the study in general. There was no verbal feedback collected.

OK, then I misunderstood this.

Author Response

Dear Reviewer:

Thanks for your comments. Please find our reply under the three main points which are made considering all your comments to our earlier responses below.

Three minor remarks I still have:

  1. present your implementation if the IVA much earlier in the motivation and add the motivation about the relevance of your study

RESPONSE:

  • Sorry for the confusion regarding the used terms. You are correct, maybe the use of the term “embodied conversational agent” was better than “intelligent virtual agent”.

We have modified the second sentence in the first paragraph of the introduction from:

To build acceptable and engaging IVAs towards fruitful interactions, the design of IVAs requires multidisciplinary effort including psychology [4] and artificial intelligence [5] expertise to achieve capabilities such as emotion modeling [6,7].”

To:

Compared to conversational agents (like chatbots or voice assistants) that do not have a visual representation or embodiment, building acceptable and engaging IVAs towards fruitful interactions is more challenging due to the appearance dimension included in the design of IVAs. For this reason, they are also commonly known as embodied conversational agents. The design of IVAs requires multidisciplinary effort including psychology [4] and artificial intelligence [5] expertise to achieve capabilities such as emotion modeling [6,7].

  • We added figure 1 to present the IVA and illustrate how the application appears at the user end (frontend).
  • We added the motivation (as described below with some edits) to the last paragraph of the introduction.

  1. give some more technical details (e.g. FAtiMA and the system you used to generate the TTS-voice.

RESPONSE: Sorry again for missing this point. We have added more details in the second paragraph of Section 3.1. study design and materials, as follows:

“Second, for the cognitive side of the agent that controls its behaviour, we utilised the cognitive agent architecture FAtiMA (Fearnot AffecTIve Mind Architecture) [54]. More details on how we used and modified FAtiMA in our work can be found in [56]. The FAtiMA toolkit allows us to manage the agent-user dialogue by controlling the transition/turn-taking between the agent and the user according to the user’s response. For example, as illustrated in Figure 1, based on the user’s selection, the agent’s response varies. Once the dialogue is scripted, the toolkit generates the audio files for all the agent’s utterances using TTS with any voice of choice (e.g., the voices available in Microsoft windows language packs).”

Further, we have replaced the name “UK English Hazel” with “Microsoft Hazel the UK English voice

  1. justify why mimikry is important for IVAs

The concept of mimicry in our article is centred around anthropomorphism. We have sought to justify why anthropomorphism is sought after in IVAs. Our only mention of mimicry is use of the term mimic in the opening sentence “designed to mimic the natural human-human interaction”. Similarly, we use the term, simulation to talk about the IVA naturally simulating humans. We defined this as anthropomorphism. Thus mimicry, simulation and anthropomorphism all refer to making the IVA humanlike. As stated in the sentence below, anthropomorphism aims to increase acceptance:

“Great effort is being devoted to increase this acceptance by giving the technology life by simulating the natural look and behaviour of a human being (i.e. anthropomorphism) [11].”

In answer to the question “Especially how are enhancing co-presence and likability/voice-impression related to mimikry?”

After discussing the role of social presence and co-presence in human-human context, we further discuss these in an IVA context by considering anthropomorphism and its claimed benefits as in the following sentences:

“This differentiation between social presence and co-presence is important to understand the impact of anthropomorphism. In the literature, it is assumed that increasing the agent’s realism (anthropomorphism) leads to higher social presence, and consequently, increases the IVA’s likeability, and acceptance (e.g., [14,15]) with contradictory findings regarding the impact of anthropomorphism on the user-agent relationship and interaction goal (e.g., [16-19]).”

Please advise if this is sufficient clarification.

Further regarding your comment on the study design: I’m not fully convinced by that, because possible carryover impressions could be avoided by having a random order of the experiments for all participants. But I know that it is hard to change a once conducted experiment, so maybe just add the aim to avoid carryover effects.

RESPONSE: We have added the following statement in the first paragraph of the methodology:

“Due to potential carryover effects, we chose not to expose participants to both voice settings.”

Thanks for your feedback.

The paper describes a study to decide whether TTS voices or human voices are better for IVAs to facilitate so called co-presence.

I have several concerns with this study:

  1. a) The introduction is quite broad and a lot of concepts and terms are used without a further explanation. For example:

-That the authors are working with an avatar-based system instead of an interactive voice assistant (that maybe do not have a visual presence) is just knowable from later sections.

RESPONSE: We commenced our article with a definition of the term we use throughout, including in the title, to refer to the type of technology we are using: intelligent virtual agents. We are not using a voice assistant and we do not use that term anywhere. We are also not using an avatar-based system. The word avatar is not used once in our article. We are using an agent-based system, hence the use of the term “agent” and also the use of the term “intelligent”. We have not created this term. This term is widely known and used to refer to the AI-driven agents with visual appearance.

When we use the term agent, as used by IVA researchers, we refer to a computer-based agent that is artificially intelligent, as indicated by the term “intelligent”. To make this more apparent to readers not familiar with IVA technology or research in this field, we have added the words “artificially intelligent” to the definition in our opening sentence. Our description “animated characters, designed to mimic the natural human-human interaction” sought to convey it has a visual representation/embodiment. The other term commonly used for this technology is Embodied Conversational Agents, but we use the more general term used by the community. We chose not to introduce any further terms to avoid confusion. We could add the word “embodied” before “animated character”, but that seems to be redundant.

I know that the term IVA is widely known and used, but there are different implementations. Researchers comprise voice assistants (i.e. smart speakers) as well as chatbots and agents having a virtual (human-like) representation (i.e. Max from uni bielefeld) as a IVA. Therefore, the term ECA (embodied conversational Agent) is used to specify a sub-group of IVAs.

Therefore, and to understand the experimental setting, this should be made clear in the beginning of your manuscript. Thus, I recommend to define your implementation of an IVA in the beginning if the introduction. Furthermore, a scree-shot of the agent would be good in the Methods chapter

-why should IVAs mimic natural human-human interactions?

RESPONSE: Mimicking human-human interaction or “anthropomorphism” is an ongoing research area. The literature is rich with findings on how more humanlike features (e.g., look, sound, body gesture, behaviour, etc) embedded in IVAs could improve the user experience interacting with the IVA and could reap better system outcomes. For more information, we have cited a recent review ([15] in the paper) which reviewed the literature of anthropomorphism in the field of IVA.

Because the anthropomorphism topic is very broad, in our paper, we only focused on the findings of anthropomorphism and the related terms in the paper, for examples, social presence, likeability, and congruence.

I’m not asking for a definition of mimikry, I was asking to give a justification why a ECA should do that, what are the assumptions regarding the dialog? Especially how are enhancing co-presence and likability/voice-impression related to mimikry?

-what are believable IVAs?

RESPONSE: the notion of “believable agent” is widely used in the field. The simplest definition is “..provides the illusion of life..” by Joseph Bates in “The role of emotions in believable agents”. IVA researchers use the term ‘believable’ to describe the agent’s human-like features such as believable emotion, behaviour, personality, and so on. Please note that, in our paper, we didn’t use this definition and we did not seek to measure believability. To avoid any confusion and decrease the definitions, we replaced the word ‘believable’ with ‘acceptable and engaging’.

Thanks!

-what is the eeriness problem (not introduced in the introduction)

RESPONSE: Eeriness problem is an alternative term to “the uncanny valley”. We added the term “uncanny valley” after the term “eeriness problem” in the paper:

However, considering the uncanny valley problem, Brenton, et al. [20] suggested that the eeriness problem or “uncanny valley” occurs as a result of how the user perceived the agent’s co-presence.”

Thanks!

-The statement Co-presence is defined as having... (line 55ff) is missing a reference.

RESPONSE: We added the reference.

Thanks!

line 123: In this study -> in the actual study

RESPONSE: We changed “In this study” to “In our study reported this paper,..” to distinguish between “the recent study” mentioned in the previous sentence and “our study”.

OK.

The methods chapter leave the reader alone with a lot of design decisions the authors made:

  1. why is just a between subjects design used and not a within-subject cross experiment (when the dialogs are just 8 minutes and some time used for the questionnairs (how much, never mentioned) there should be enough time for two rounds (random order)

RESPONSE: In line 296, we reported that “Participants were randomly assigned to interact with VA-recorded or the VA-TTS.” Hence, we claimed that our experiment is of between-subjects design. Participants did not interact with both virtual agents.

We reported an average of 8 minutes, but some participants took longer. We have up to 30 minutes for the study with all of the pre and post questionnaires. We have added the sentence “Participants took between 20-30 minutes to complete the entire study.” In line 305, modified version of the paper.

In addition to not having time for both a within and between subject design, we did not want to contaminate/bias the second interaction due to possible carryover of impressions from the first round. This design kept our statistical analyses simpler.

I’m not fully convinced by that, because possible carryover impressions could be avoided by having a random order of the experiments for all participants. But I know that it is hard to change a once conducted experiment, so maybe just add the aim to avoid carryover effects.

  1. Why do you use a female voice, does that have an influence on the perception?

RESPONSE: We have added the following statement to Section 3.1. Study Design and Materials.

The choice of a female VA was arbitrary as the agent’s gender has little to no influence on the user experience and participants tend to prefer younger female agent [51].

Furthermore, both the human and agent voice are female. So if there are any gender-related effect it should apply to both treatments.

In section 5, we discuss the character’s gender as part of responding to a participant’s comment. We also clarify that

However, in our analysis, between-gender statistical analysis did not reveal any differences in terms of likeability, voice eeriness, working alliance, trustworthiness (except for benevolence), or trust.”

We have now added to the end of this sentence “Nevertheless, future work could follow the participant’s suggestion to match human-agent gender to measure any impacts.”.

Thanks, that what I intended, to have an additional discussion of that point

  1. You only mention the TTS voice but not the voice SDK/API or system.

RESPONSE: We did not need to use SDK/API. Our system (using FAtiMA) uses either generated voices (TTS technology) or recorded voices. We have clarified this point in the paper as follows:

“In the second setting, VA-TTS, the utterances were generated using UK English Hazel. Although synthetic voices with higher quality have been introduced [54,55], we opted to use the basic synthesised voice of Hazel for two reasons. First, some higher quality synthesised voices could closely imitate the human voice (e.g. Sisman, et al. [56] and Shen, et al. [57])….”

I think to understand the experiment and the utilized technique (recapitulation and comparisum of experiments), it is important to get the information of the framework (FatiMA) and the TTS-voice. And I still do not get the point about the voice. You wrote you use UK English Hazel, but from which system: MaryTTS, Amazon Polly, etc? This is important, if others do similar experiments with other voices to then compare the voice impression also acoustically.

  1. What is about the relevance of your study, if you admit that there are better TTS-voices that are indistinguishable from real human voices. Wouldn't that mean that in a few years the TTS-quality does not have an influence on the co-presence at all? So why bother?

RESPONSE: This study is part of a project where we investigate different factors on building user-agent therapeutic alliance including explanation and empathy. Some of the participants in previous experiments of the project complained about the voice being robotic. We sought to investigate the impact of this factor on our main desirable outcome (trust and behaviour change), mainly, to validate our results of the project in general. We mean by validation here is to find if the type of voice is a cause of reduction in user-agent trust/alliance or in behaviour change. If there was a significant difference in our study, reported in this paper, favouring the natural/recorded voice, we should reconsider our design and previous results as well.

The same concern applies to most of the current studies in the literature that use TTS in building their chatbot or virtual agents. Our study contributes to the literature in validating others’ findings. Currently, the use of high-quality TTS is expensive to some extent or not available to some researchers.

OK, can you put that soimehow into the motivation?

  1. line 238 -> "Sara is downloadable" -> Who/What is Sara

RESPONSE: Sorry for this mistake. Sarah is the name of the VA in the experiment which we didn’t report it in the paper. We replaced the name “Sarah” with “the VA” in the paper.

OK.

  1. regarding the conversation, a sample dialog would be helpful

RESPONSE: We have published the dialogue in our previous papers which we will refer to at the publication stage. We didn’t present parts of the dialogue in this paper as our main focus is the voice type regardless the content, especially that the content of both VAs was identical.

OK.

  1. how can you guarantee that the pre-recorded utterances fit to the dialog?

RESPONSE: The dialogue is first crafted as text. Next, the text that represent the agent’s responses: 1) is used as input to the TTS generator and 2) is read by the female reader and recorded with the help of an expert using Audacity program. Every response of the agent is recorded as a separated audio file. Further, the text/response and its corresponding audio are paired programmatically for later use. In the dialogue tree (agent-user turn taking), the audio files (TTS or recorded) are used to generate the VA’s behaviour (speech): synchronizing the audio file with the VA’s lips’ movement, and their paired text are displayed on the screen as subtitles.

We can add this further description to the paper if deemed important for the reader.

Oh sorry for the confusion, I meant if the participants can answer freely, how can you guarantee that you have a suitable pre-recorded answer?

  1. The description of all the questionnaires is a bit confusing, a procedure scheme would be helpful.

RESPONSE: A new figure (Figure 1) is added to clarify the study design flow chart.

Thanks!

Comments regarding the results section:

  1. line 301 "6.6% North American and Middle East" each or both together?

RESPONSE: yes, combined. We added the word at the end of the sentence “combined” after "6.6% North American and Middle East". Please note that the culture group question is designed to include the twelve main cultures in Australia, which are defined by the Australian Bureau of Statistics (2016).

Thanks

  1. Table 1: Why are the genders so imbalanced between the two conditions?

RESPONSE: In every group, the female participants number is greater than the male. However, there was no significant difference between the two groups in terms of female to male distribution. This ingroup imbalance shouldn’t impact the results as we further ran different statistical tests to investigate if the gender (within the experimental groups) has an influence on other factors as reported at the very end of the discussion section:

“However, in our analysis, between-gender statistical analysis did not reveal any differences in terms of likeability, voice eeriness, working alliance, trustworthiness (except for benevolence), or trust.”

OK Thanks!

  1. in section 4.3. the authors should use the same terms as used in the figures, especially "boring (line 345) is not visible in Fig2!.

RESPONSE: With the new Figure 1, Figure 2 is now Figure 3. We can see the word “boring” at the bottom of Figure 3. We believe some error must have occurred in the creation of the pdf available to the reviewer.

Thanks

In general, the authors should re-think the chosen presentation forms for the results. Sometimes, they used tables, sometime line-plots, and sometimes bar plots for the same type of data. Why?

This hinders a better comparison.

RESPONSE: Sorry if our choice of presentation of results caused any confusion. We tried to simplify the results for the reader by choosing the best presentation for every type of data. While we could report all the results in tables, we used visual presentation for likeability and voice impressions to show the tendency of the participants in each group towards the two poles. However, the voice understandability data was not applicable to be presented in a similar way, so we opted to use bar graph, again to visualize the results and reduce stuffing the paper with more numbers and tables.

OK!

Furthermore, where is Fig 1 and why does Fig. 2 consists of two graphics with different aspect ratio, and the upper graphic is missing an x-axis.

RESPONSE: Sorry, the graphs overlapped at some point before the submission, although we ensured everything was in place. We fixed the problem now.

Thanks!

Discussion:

In the discussion, the authors tried to refer their results back to their research hypotheses. But it could be depicted in a better way, as actually the different hypotheses are just mentioned in the text, without rephrasing the hypothesis text. Furthermore, I would recommend discussing the actual results in comparison to other results especially regarding [22], because it is a bit strange to start with the hypothesis that TTS and human voices have an influence but then all the assumptions fail.

RESPONSE: Thanks for your suggestion.

We followed a very standard and common method of investigating the impact of an independent variable (i.e., voice type) on dependent variables (co-presence, trust, likeability,…) with hypotheses formulation based on the literature. Hence, the discussion should clearly report how the results support or do not support these hypotheses (accept or reject the hypotheses). Failing to support the hypotheses means that the alternative hypotheses are true, and consequently, there is an alternative path for the researchers/readers to follow.

The study in [22] differs from our study so we sought not to compare our results to their results. The study in [22] used virtual reality environment rather than an intelligent virtual agent that is presented on any screen. They further measure social presence rather than co-presence. However, we discussed our results against the literature in the discussion (e.g., paragraphs start at line 424 and line 444 in the modified version of the paper)

Furthermore, I only noticed after reading the discussion that the participants gave verbal feedback.

RESPONSE: The participants provided feedback at the end of the study as a reply to a free-text question about their opinion on the study in general. There was no verbal feedback collected.

OK, then I misunderstood this.
